# Learning Multilingual Expressive Speech Representation for Prosody Prediction without Parallel Data

*Jarod Duret[1], Yannick Estève[1], Titouan Parcollet[2]*

[1]LIA - Avignon Universite, France
[2]University of Cambridge, United-Kingdom
jarod.duret@univ-avignon.fr

## Abstract

We propose a method for speech-to-speech emotion-preserving translation that operates at the level of discrete speech units. Our approach relies on the use of multilingual emotion embedding that can capture affective information in a language-independent manner. We show that this embedding can be used to predict the pitch and duration of speech units in a target language, allowing us to resynthesize the source speech signal with the same emotional content. We evaluate our approach to English and French speech signals and show that it outperforms a baseline method that does not use emotional information, including when the emotion embedding is extracted from a different language. Even if this preliminary study does not address directly the machine translation issue, our results demonstrate the effectiveness of our approach for cross-lingual emotion preservation in the context of speech resynthesis.

**Index Terms**: speech synthesis, prosody prediction, speech generation

## 1. Introduction

Speech-to-speech translation has become increasingly important in today's globalized world, facilitating communication across different languages and cultures. However, current speech-to-speech translation systems often fail to preserve the emotional nuances of the speaker's original message, which can lead to misinterpretation and misunderstandings. Emotion is a critical aspect of human communication, and preserving it in speech-to-speech translation can greatly enhance the accuracy and effectiveness of the translation.

Recently, a textless direct speech-to-speech translation (S2ST) approach has been proposed that is based on the use of discrete speech units [1]. Such an approach is particularly interesting to translate from an unwritten language and/or to an unwritten language. It is also noticeable that it is very efficient to process speech-to-speech translation for languages for which a written form exists [1, 2]. By utilizing this method, we aim to improve the accuracy of speech-to-speech translation models in capturing the linguistic content of the target speech without being influenced by the speaker's prosodic features. Indeed, these discrete speech units can disentangle linguistic content from prosodic and speaker identity information in speech, as shown in [3]. Nevertheless, this loss of prosodic information is an issue to preserve the expression and the emotion of the source utterance: prosody is an essential aspect of speech and can convey important emotions and attitudes. Another issue to achieve emotion preservation in speech-to-speech translation is the lack of parallel speech data necessary to train systems.

The work presented in this paper is the first step of our study toward emotion preservation in speech-to-speech translation. Inspired by recent work on textless speech emotion conversion using discrete representations [4], we propose an approach that aims to build a speech-to-speech translation – based on the use of discrete speech units – that preserves the emotion, without the need for parallel speech data.

Our approach is based on the use of both Pitch Predictor and Duration Predictor models. These models are conditioned by an emotion embedding and a sequence of discrete speech units. The emotion embedding is a continuous vectorial representation in a space shared by different languages. In this paper, we show how this emotion embedding is constructed on different monolingual emotion datasets dedicated to different European languages. While we do not address the translation module in this line of research, our experiments show that this emotion embedding allows us to preserve the emotion despite the reduction of the speech signal in a sequence of discrete speech units. Our experiments on the prosody reconstruction are made on both a monolingual and a bilingual scenario.

## 2. Related work

**Multilingual Speech Emotion Recognition**. In recent years, neural-based models have started to replace traditional machine learning approaches [5][6] in Speech Emotion Recognition (SER). Convolutional neural networks (CNNs) and recurrent neural networks (RNNs) have been shown to achieve better performance in emotion recognition tasks [7][8]. More recently, transfer learning has also been explored in the field of speech emotion recognition, with self-supervised pretraining being a popular approach. Models such as wav2vec [9] and HuBERT [10] have shown state-of-the-art performance in speech emotion recognition [11, 12, 13].
In addition to monolingual SER, recent works have started to explore SER in a multilingual context. A common approach for achieving multilingual training is to merge the available data in each language into one corpus and train them together. It is also possible to train the model on one language and then fine-tune it on other languages as shown in [14, 15].

**Spoken Language Modeling from audio**. Generative Spoken Language Modeling from audio refers to the task of learning the acoustic and linguistic characteristics of a language from the raw audio i.e. no text and no labels. In [10], the authors proposed utilizing the recent advances in self-supervised speech representation learning to discover discrete speech units and model them. Then, they showed that speech generation can be accomplished by sampling unit sequences from a unit-discovery model and synthesizing them into speech waveform by using a unit-to-speech model. More recently, [3] demonstrates the effectiveness of using self-supervised learned

discrete speech units for high-quality speech synthesis, without the need for Mel-spectrogram estimation. Lastly, a similar approach and speech representation scheme was utilized by [4] for textless speech emotion conversion via translation. Here, we leverage a similar approach but in a multilingual context.

**Direct speech-to-speech translation**. In [16], the authors proposed an attention-based sequence-to-sequence neural network that can directly translate speech without using an intermediate text representation. The model was trained end-to-end to map speech spectrograms into target spectrograms in another language. Additionally, the authors demonstrated a variation that simultaneously transfers the voice of the source speaker to the translated speech. However, the authors reported that the voice transfer did not perform as well as in a similar text-to-speech context, which highlights the challenges of the cross-language voice transfer task. Then, [1] introduces a direct S2ST system based on self-supervised discrete representations. According to the authors, the system outperforms the VQVAE-based approach in [17] when trained without text data. Lastly, [2] tackles direct S2ST by following [1] and focuses on training the system with real data. The authors also introduce a self-supervised unit-based speech normalization technique which reduces variations in the multi-speaker target speech while retaining the lexical content. Prior research in direct S2ST has been primarily focused on the translation of linguistic content while not considering the paralinguistic aspect. In contrast, we aim to build a similar approach that preserves emotion. Also, we do not address the translation aspect and focus on the prosody generation related to a sequence of discrete speech units.

## 3. Method

The proposed architecture is comprised of two pre-trained fixed encoders, namely: (i) speech content encoder; (ii) emotion encoder; and three independent models, namely: (iii) Duration Predictor; (iv) Pitch Predictor; (v) speech vocoder. The first encoder constructs a sequence of discrete speech units by extracting a sequence of continuous representations on which quantization is applied. The second encoder generates an emotion embedding. Details about these encoders are described in subsection 3.1, details about the duration and Pitch Predictors and the vocoder are presented in subsection 3.2 while the overall architecture is illustrated in Figure 1.

### 3.1. Encoders

**Phonetic-content Encoder** To represent the linguistic content (mainly pseudo-phonetic information) contained in speech, we extract raw speech features from the audio signal using a pre-trained SSL model, namely HuBERT [18] for English and a multilingual HuBERT (mHuBERT) [2] for French. We chose these two models since they were shown to better disentangle between phonetic-based content and both speaker and prosody compared to other SSL-based models [3]. Since the representations learned by HuBERT and mHuBERT are continuous, a $k$-means clustering is performed on the features and the learned $K$ cluster centroids are used to transform audio into a sequence of cluster indices at every 20ms. For the rest of the paper, we refer to these units as "discrete units". For English speech, we extracted representations from the $6^{th}$ layer of HuBERT model and set $k = 100$ as used in [1] for speech-to-speech translation. For French speech, we have been following [2] and extracting representations from the $11^{th}$ layer of mHuBERT

Figure 1: *An illustration of the proposed system. First, the input signal is encoded as a sequence of discrete units. Next, we predict the duration and F0 before feeding them to a unit-to-speech model. Duration Predictor, Pitch Predictor and the unit-to-speech model are conditioned by the emotion embedding extracted from the emotion encoder. The speaker is encoded using a 1-hot vector directly in the unit-to-speech model.*

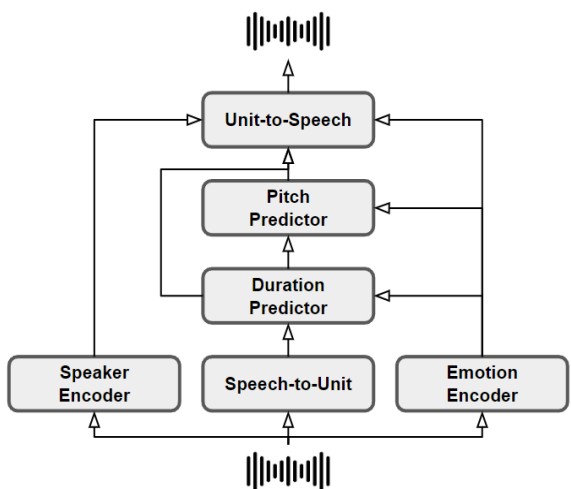

model and set $K = 100$. Following [1][4], we have experimented with reducing a sequence of units to a sequence of unique units by removing consecutive duplicated units (e.g., 0, 0, 1, 1, 1, 2 → 0, 1, 2). We denote such sequences as "reduced".

**Emotion Encoder** In order to create representations of emotion, a distinct encoder was trained for emotion recognition tasks within a multilingual framework. Our proposed architecture consists of an upstream encoder, Wav2Vec2-XLSR [19], that has been pre-trained through semi-supervised learning, a bottleneck layer, and a dense layer with a softmax activation that returns probabilities for each of the emotion classes. Following the approach outlined in [13], both the CNN and Transformer modules of the Wav2Vec2-XLSR model were fine-tuned during the downstream model training process. We experimented with multiple SSL models as encoders and report the results in Table 1. In a subsequent step, we employed the previously trained model as an encoder to generate a continuous vectorial representation of dimension 96 for each utterance. This process was accomplished via the bottleneck layer, followed by temporal pooling to reduce the information along the entire speech sequence into a single vector representation. These embeddings were then employed to condition both a Pitch Predictor model and a Duration Predictor model.

**Speaker Encoder** Lastly, we construct a speaker representation included as an additional conditioner during the speech synthesis phase. This can be accomplished by using a pre-trained speaker verification model similar to the one proposed in [20] or by optimizing the parameters of a fixed size look-up-table. We experimented with the second option even though using speaker representation coming from a pre-trained model gives the ability to generalize to new and unseen speakers, it is slightly less efficient for capturing only information related to the speaker identity as observed in [4].

### 3.2. Models

**Duration Predictor** As a result of the use of reduced sequences, we first have to predict the duration of each discrete speech unit. For this purpose, we use a CNN to learn the correspondence between the phonetic-like content units and their duration. Our Duration Predictor is adapted from [21] where the Duration Predictor takes the phoneme hidden sequence as input and predicts the duration of each phoneme. In this work, we replace the phoneme sequence with the reduced discrete unit sequence and predict the number of repetitions for each unit. Following [4], the model is also conditioned by emotion embedding. During training, we use the ground-truth discrete unit durations as supervision and optimize the model with mean square error (MSE) loss.

**Pitch Predictor** We follow [4], and use a F0 estimation model to predict the pitch from a sequence of speech discrete units along with an emotion embedding. The model is a CNN followed by a linear layer projecting the output to $\mathbb{R}^d$. A sigmoid is applied on top of the network to output a vector in $[0, 1]^d$. During training, the target F0 is extracted using the YAAPT [22] algorithm. Ranges of F0 values are discretized into $d$ bins that are represented by one-hot encodings. To allow a better output range when converting bins back to F0 values, we output the weighted average of the activated bins. Finally, we use the mean and standard deviation for each speaker to standardize the F0 values.

**Speech synthesis** For the unit-to-speech model, we use the HiFi-GAN neural vocoder [23]. HiFiGAN is a generative adversarial network (GAN), it consists of one generator and two discriminators: multi-scale and multi-period discriminators. The generator is a fully convolutional neural network, we adapted the encoder part following [3] to take as input a sequence of discrete-unit, predicted F0, emotion-embedding, and speaker-embedding. Before feeding the above features to the model, we concatenate them along the temporal axis, then we apply a linear interpolation to match the sample rates of the unit sequence with the F0. Speaker-embedding and emotion-label are replicated along the temporal axis.

## 4. Experiments

This section validates our model following a two-step process. First, we must ensure the validity of our multilingual emotion embedding (Section 4.1). Second, and building on this primary analysis, we perform and evaluate our speech synthesis both from quality and emotional points of view (Section 4.2).

### 4.1. Multilingual emotion recognition

#### 4.1.1. Experimental setup

We are considering only datasets annotated with emotion, speaker, and sex labels. We selected four emotional labels (Neutral, Angry, Happy, Sad), and only samples annotated with these labels are extracted from the datasets. Indeed, only these labels appear in all the databases listed below. Only European languages are considered.

**IEMOCAP** [24] is an acted, multimodal and multispeaker database. It contains 12 hours of speech from 10 English speakers (5F, 5M).

**CREMA-D** [25] is a crowd-sourced emotional multimodal acted database. It contains 7,442 English clips from 91 speakers (43F, 48M). These clips were recorded by actors between the ages of 20 and 74 coming from a variety of races and ethnicities (African America, Asian, Caucasian, Hispanic, and Unspecified).

**ESD** [26] is a multilingual emotional database. It consists of 350 parallel utterances spoken by 10 native English and 10 native Chinese speakers (10F, 10M). We only consider the English part.

**Synpaflex** [27] is an expressive French audiobooks database. It contains 87 hours of speech, recorded by a single speaker. The subset annotated with emotional labels contains 8,751 clips.

**Oreau** [28] is a French emotional speech database. It contains 502 utterances from 32 non-professional actors (7F, 25M).

**EmoDB** [29] is a German emotional speech database. It contains a total of 535 utterances from 10 professional speakers (5F, 5F).

**EMOVO** [30] is an Italian speech database. It contains 14 sentences over seven different emotions from 6 professional actors (3F, 3M).

**emoUERJ** [31] is a Portuguese emotional speech database. It contains 377 utterances from 4 professional speakers (3F, 3F).

We converted every dataset to 16-bit PCM with a sample rate of 16 kHz. The data were separated into two distinct sets with no overlap using an 80% ratio for training and a 20% ratio for evaluation and testing. Emotion classes were balanced across each language in for each set.

#### 4.1.2. Results

In this first experiment, we aim to learn a multilingual speech representation by exploring various model architectures. To achieve this, we utilize self-supervised learning (SSL) pre-training and fine-tuning approaches with four different systems, namely HuBERT, Wav2Vec2, mHuBERT, and Wav2Vec2-XLSR, on a multilingual Speech Emotion Recognition (SER) task. As a baseline comparison, we also employ a CNN model using Fbank as input. According to what we found in the literature, there is no previous work that is comparable to our experiment. It is important to note that the aim of this study is not to develop a state-of-the-art multilingual SER system, but rather to investigate the effectiveness of the proposed approach to learning a multilingual speech emotion representation. We report the performance results of our experiments in Table 1. We can see that Wav2Vec2-XLSR outperforms or performs comparably well to other architectures for French, German, and Portuguese, while the difference in performance is marginal for English. Based on this observation and our focus on English and French languages, we choose Wav2Vec2-XLSR as the default architecture for subsequent experiments.

Then, in this second experiment, we evaluated the performance of the Wav2Vec2-XLSR model in a cross-lingual setting. To be exact, we independently fine-tune the model for each language and then evaluate its performance in all available languages. This evaluation was motivated by the limited

Table 1: *Results from systems trained on multilingual emotion recognition task. Average accuracy (%) obtained for the different models and languages.*

| Models | Language | | | | |
|---|---|---|---|---|---|
| | English | French | German | Italian | Portuguese |
| Fbank | 0.84 | 0.56 | 0.81 | 0.81 | 0.77 |
| HuBERT | **0.93** | 0.67 | 0.94 | **0.96** | 0.95 |
| Wav2Vec2 | 0.92 | 0.69 | **0.97** | 0.92 | 0.95 |
| mHuBERT | 0.91 | 0.68 | 0.92 | 0.94 | 0.92 |
| Wav2Vec2-XLSR | 0.92 | **0.74** | **0.97** | 0.92 | **0.97** |

availability of data for languages other than English. Therefore, we aim to explore the potential benefits of multilingual training on model performance. The obtained results are reported in Table 2. The results indicate that the multilingual setup outperforms the monolingual one for all selected European languages, except English by a very minor margin (0.92% of accuracy instead of 0.93%). This highlights the potential benefits of incorporating multilingual data during training, particularly when the amount of available data for a specific language is very limited.

Table 2: *Results from systems trained on cross-lingual emotion recognition task. Average accuracy (%) obtained by training the system independently for each language and evaluate its performance on all available languages. For each models we use the Wav2Vec2-XLSR as the default architecture.*

| Models | Language | | | | |
|---|---|---|---|---|---|
| | English | French | German | Italian | Portuguese |
| English | **0.93** | 0.46 | 0.94 | 0.67 | 0.88 |
| French | 0.61 | 0.70 | 0.64 | 0.50 | 0.67 |
| German | 0.44 | 0.29 | 0.94 | 0.64 | 0.77 |
| Italian | 0.39 | 0.26 | 0.50 | 0.89 | 0.50 |
| Portuguese | 0.45 | 0.32 | 0.64 | 0.39 | 0.92 |
| Multilingual | 0.92 | **0.74** | **0.97** | **0.92** | **0.97** |

Next, we conducted an analysis of the embeddings generated by the emotion encoder on a subset of the test corpus. We randomly selected four speakers for each available language and sampled one utterance for each of the emotional labels considered. Specifically, we applied a $k$-means clustering with $k = 4$ (the number of emotion labels) on top of the emotion embeddings. To evaluate the quality of the clustering in terms of emotional states, we calculated the V-measure [32], an external entropy-based cluster evaluation measure. The V-measure returns a score between 0.0 and 1.0, with 1.0 indicating perfect labeling. Using the Wav2Vec2-XLSR model, we obtained a V-measure score of 0.76, indicating a reasonably good clustering performance of the emotions. This suggests that the emotion encoder can capture the emotional features of the input speech signal, regardless of the language spoken, and that these features can be efficiently separated into the corresponding emotion categories by the $k$-Means algorithm.

Finally, we present a two-dimensional visualization of the emotion embedding space utilizing the t-SNE algorithm in Figure 2. This visualization provides a useful representation of the high-dimensional multilingual emotion embedding space and can provide a better understanding of the relationships between different emotions and languages in this space. The observed

Figure 2: *Visualization of emotion embedding using t-SNE algorithm. In this visualization, each data point corresponds to a distinct utterance, where the emotion label is depicted by distinct colors, and the language is represented via various shapes*

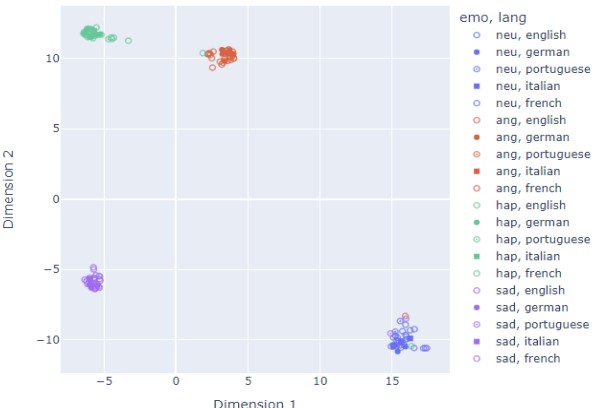

embeddings exhibit clear separation into four distinct clusters, each corresponding to an individual emotional class, irrespective of the language.

### 4.2. Speech synthesis

To train our unit-to-speech system for English language, we utilize the LJSpeech dataset [33], which is widely used for training speech synthesis models. The LJSpeech dataset contains 13,100 short audio clips of a single speaker reading passages from 7 non-fiction books, with a total duration of approximately 24 hours. In addition to the LJSpeech dataset, we utilize the English portion of the ESD dataset for training and evaluating our unit-to-speech model. For French language, we train our model on the Synpaflex dataset which is a single-speaker corpus. To evaluate the performance of our model, we conducted experiments using the Synpaflex and Oreau datasets.

#### 4.2.1. Results

We evaluate the pitch reconstruction of our approach on the test set by computing the concordance correlation coefficient score [34](CCC) on the F0 extracted after synthesis. CCC is defined as follows:

$$\rho_c = \frac{2\rho\sigma_x\sigma_y}{\sigma_x^2 + \sigma_y^2 + (\mu_x - \mu_y)^2} \quad (1)$$

where $\mu_x$ and $\mu_y$ are the means for the two variables, $\sigma_x^2$ and $\sigma_y^2$ are the corresponding variances. $\rho$ is referring to the correlation coefficient between the two variables (in our case the pitch values in the natural speech audio file and the values of the resynthesized speech).

In this experiment, we aim to resynthesize an utterance by keeping the prosody after having converted the natural speech signal into a sequence of discrete speech units. As presented in subsection 3.1, we use 100 different discrete units. Our objective is to evaluate the capability of our multilingual emotion embedding to provide relevant information to the pitch and Duration Predictors used for resynthesis. As part of a contrastive

experiment in our evaluation setup, we directly extract the pitch and duration information from the speech signal and then provide this data, along with the corresponding sequence of discrete speech units, to the unit-to-speech model. This approach yields the best possible resynthesis of the speech signal, and we refer to it as "Oracle". We compare this to a baseline approach in the second row of 3 and 4, which relies only on the sequence of discrete speech units. Then, we assess our proposed approach, which predicts F0 and duration from the sequence of discrete speech units and an emotion embedding.

Since this study is the first step toward a speech-to-speech translation that preserves the source emotion, we finally conduct a cross-lingual experiment by randomly selecting a French utterance from the Oreau dataset with the same emotion label and extracting the emotion embedding from it. We repeated five times the cross-lingual experiment, and the average CCC is reported. We can see that our proposed approach outperforms the baseline even when conditioning the Pitch Predictor and Duration Predictor with a representation from another language. We conducted the same experiment in the opposite direction, results are reported in Table 3 and Table 4. In summary, the experimental results demonstrate the effectiveness of the proposed approach in leveraging cross-lingual information for prosody prediction.

Table 3: *Concordance Correlation Coefficient (CCC) score computed on F0 extracted from English speech*

| Emotion | F0 | Neutral | Happy | Sad | Angry |
|---------|----|---------|-------|-----|-------|
| Oracle  |    | 0.67    | 0.83  | 0.80 | 0.81 |
| -       | -  | 0.49    | 0.45  | 0.54 | 0.53 |
| EN      | EN | 0.59    | 0.60  | 0.66 | 0.59 |
| FR      | EN | 0.51    | 0.61  | 0.63 | 0.60 |

Table 4: *Concordance Correlation Coefficient (CCC) score computed on F0 extracted from French speech*

| Emotion | F0 | Neutral | Happy | Sad | Angry |
|---------|----|---------|-------|-----|-------|
| Oracle  |    | 0.72    | 0.87  | 0.64 | 0.83 |
| -       | -  | 0.52    | 0.57  | 0.44 | 0.51 |
| FR      | FR | 0.61    | 0.71  | 0.48 | 0.68 |
| EN      | FR | 0.56    | 0.68  | 0.47 | 0.59 |

In addition, we performed an automatic evaluation to determine if our proposed approach can preserve the emotion of the original speech despite the reduction of the signal to a sequence of discrete speech units. We trained a separate SER model only on English and evaluated its accuracy score using different setups. In the first evaluation setup, which we refer to as "Original", we use the reference audio recording. We compare this to a baseline approach, which relies only on the sequence of discrete speech units. Finally, we assess our proposed approach, which predicts F0 and duration from the sequence of discrete speech units and an emotion embedding. We reported confusion matrices for each of the evaluated setups.

The performance of the SER model on the speech generated by the baseline system is unsatisfactory. The model consistently

tends to predict the "angry" label, and an immediate explanation for this phenomenon cannot be deduced direclty from the analysis of Figure 3. However, it may be related to the degradation of the speech quality when only a sequence of discrete speech units is used to reconstruct the signal. When comparing the results with those obtained from the original speech, our proposed approach exhibits similar performance for the "angry" and "neutral" labels. However, a decline in accuracy is observed for the "happy" and "sad" labels. Nevertheless, this evaluation confirms the preservation of emotional content in the synthesized speech signal using our proposed approach.

To complete our analysis, we evaluated the quality of the synthesized speech using the Mean Opinion Score (MOS) metric. 15 evaluators were asked to provide a rating of the quality and intelligibility of the speech on a scale ranging from 1 to 5, where 1 represents poor quality and intelligibility, while 5 represents excellent quality and intelligibility. This assessment was designed to provide an objective measure of the overall quality of the synthesized speech: results are depicted in Figure 4. The results demonstrate that our proposed approach outperforms the baseline system in terms of Mean Opinion Score (MOS) for both English and French languages. Specifically, our approach achieved a MOS score of $3.14$ for English, which is notably higher than the MOS score of $1.75$ obtained by the baseline system. Similarly, for French, our system achieved a MOS score of $3.04$, compared to the baseline system's MOS score of $2.69$. It is worth noting that the nature of the training data used for the French language is different from that of the English language. The baseline system is performing better for the French language: this can be attributed in part to the fact that the unit-to-speech model for French is trained on a dataset with only one speaker, due to limited data availability. Nonetheless, our approach still outperforms the baseline system for both languages, indicating the effectiveness of our proposed approach.

## 5. Conclusions

By conditioning both a Pitch Predictor model and a Duration Predictor model using a multilingual emotion embedding trained on accessible datasets, we present an approach that can preserve the emotion from speech despite the reduction of the signal in discrete representations. Our experimental results, evaluated on both objective and subjective metrics, show the promises of this approach we aim to apply to speech-to-speech emotion-preserving translation. A crucial point in our contribution is that no spoken parallel data is needed to train a machine translation model that preserves emotion.

In future work, exploring alternate strategies to model prosody is an interesting direction. In the short term, we will experiment with our approach within a complete textless speech-to-speech emotion-preserving translation.

## 6. Acknowledgements

This work received funding from the European SELMA project (grant N°957017).

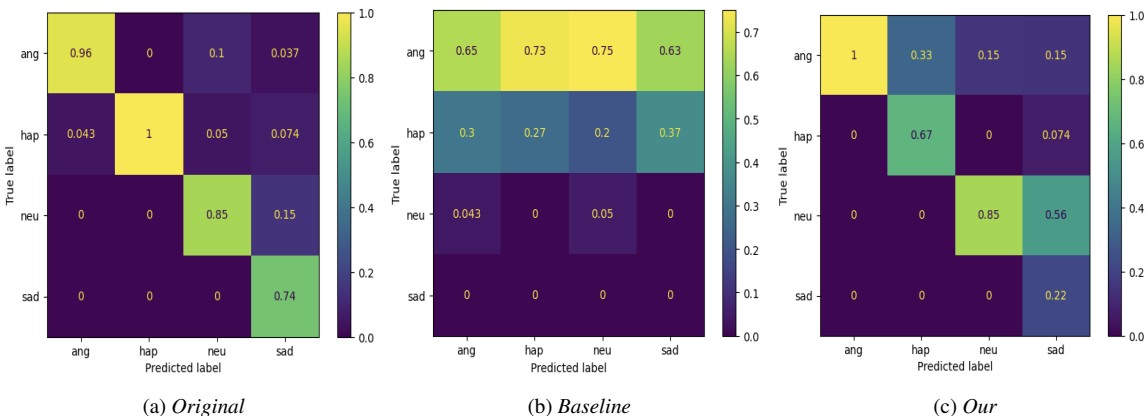

(a) *Original*          (b) *Baseline*          (c) *Our*

Figure 3: *Confusion Matrix of SER model applied to English samples generated using different setups.*

Figure 4: *Results of the objective evaluation using Mean Opinion Score (MOS) on English and French. MOS is reported using mean scores with a confidence interval of* 95%.

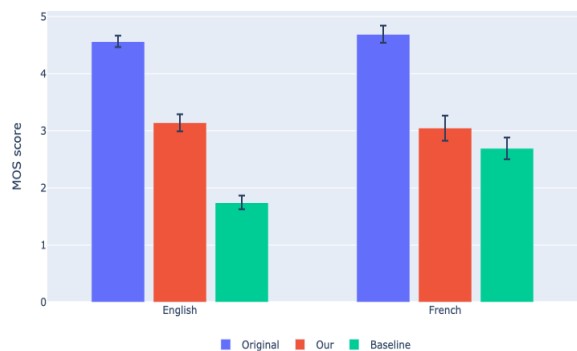

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
