# OpenReview forum: "Learning Multilingual Expressive Speech Representation for Prosody Prediction without Parallel Data"
_Interspeech.org/2023/Workshop/SSW — SSW12_

### Official Review · Reviewer_PJMC · 2023-05-29
**Preliminary study on cross-lingual emotion transfer**

**Rating:** 4
**Confidence:** 4

**Review:**

The paper presents a cross-lingual emotion transfer method that will be applied in a speech to speech translation framework in the future. By the moment, only preliminary experiments on emotion recognition and TTS are presented. While the introduction and related work sections are very clear, the experiment section needs improvement.

All the used databases are described with detail, but then many essential data are not provided. How many hours of recordings have been used for each language? What part of the data is used for training, validation and test? How have all these databases been used? Is data balanced across emotions?

Are the classification experiments been done applying cross-validation? If not, the values presented in the tables must be handled with great care, as the specific data distribution used may have influence in the results. If it has been applied (and it should have been), please add the details as how the cross-validation process has been applied.

How are the values in Tables 1 and 2 been calculated? Have these accuracy values been macro-averaged (they should have) or micro-averaged? Are the differences between the results of the systems statistically significant?
In the second emotion recognition experiment which is the model that is fine-tuned? Is it the multilingual model trained in experiment 1, or is it a generic Wav2Vec2-XLSR obtained from elsewhere? If it is the multilingual model, what is the reason to have worse results for the models fine-tuned to the language than to the multilingual model without fine-tuning for all the languages except for English? Fine-tuning should improve results, not deteriorate them.

No comment about Figure 2 is provided. If indeed the figure produces a useful representation (and I agree it does) the main conclusions that can be extracted from it should be commented in the text, and not let the task to the reader. Have you analyzed the outliers in figure 2? Is there a reason to justify them?

The first sentence of section 4.2 says that LJSpeech has been used to train unit to speech systems in both English and French, but this database contains only English recordings.

The subjective MOS evaluation asked about the opinion of evaluators on quality and intelligibility of the signals. It seems that only one questions has been asked to evaluate two very different dimensions of synthetic speech and this is not a good practice. Intelligibility is not an opinion, it can be evaluated by making the evaluators write what they understand and the calculating some metric like WER. Quality is correctly evaluated by MOS, but it does not give information about the degree of preservation of emotion. In the conclusion section, the authors say that their approach can preserve the emotion from speech but it is not possible to conclude this from the subjective evaluation as this question was not asked to the evaluators. Besides, the claim that the proposed approach outperforms the baseline system in terms of MOS for both English and French seems to be true only for English. In the French results differences do not seem statistically significant. A test to verify if this is the case should have been done (also for results presented in Tables 3 and 4).

Many references such as [9], [10], [11], [12], [13], [14], [16], [17], [19], [21], [23], lack important data as the publication conference or journal (or at least the arXiv reference if it has not been published elsewhere).

---

### Official Review · Reviewer_SXN2 · 2023-06-08
**Review - a speech-to-speech translation system preserving emotional content**

**Rating:** 6
**Confidence:** 4

**Review:**

 This study focuses on speech-to-speech translation and more specifically on emotion preservation. The proposed approach is based on a decomposition of the source signal into a sequence of discrete units using SSL techniques (as in the GSLM approach), a speaker encoder (as extended by Polyak et al.), a pitch and duration predictor (as in FastSpeech2) and, above all, an emotion encoder whose output directly conditions the synthesis. This study does not directly address the problem of translation (since the actual sequence of units is provided at test time), but rather reports preliminary experiments evaluating how the use of such an encoder can preserve emotional content in an analysis-synthesis paradigm.

   Key Strength of the paper

The proposed approach is promising, and it is interesting to begin by evaluating the impact of emotion integration on synthesis independently of the translation problem.

   Main Weakness of the paper

My main concerns relate to subjective assessment and the choice of a MOS test. First, it seems that participants must simultaneously assess quality and intelligibility, which seem to be two "orthogonal" dimensions (if quality is associated with prosodic content, then it is linked to the supra-segmental level, whereas intelligibility is generally associated with the segmental level). Secondly, I fail to see why this MOS test "indicates the effectiveness of (...) the proposed approach"? The fact that the proposed approach performs better than the baseline  shows that providing to the synthesis module an additional prosodic information (via the emotion embedding) has an impact on the overall quality of synthesis. However, this does not demonstrate that the emotional content of the synthesized speech signal is preserved (or unaffected by emotion embedding), which was the aim of the proposed study (an alternative could be to evaluate more precisely the emotional content of the resynthesized speech signal using a SER system or a (perceptual) identification test).

And also :
  - Please provide how the dataset were partitioned between train/validation/test.
- Clarify the notation «-«   in Table 3 and 4 (is it the baseline?).
- correct mathematical notation in the definition of CCC (X—>x, Y—>y).
 - Fig2 is almost not commented (consider removing it?)

   Clarity of Presentation

  English can be improved but the paper is understandable  .

Quality of References

The reference list SHOULD be cleaned  (among many issues, no conference name for ref 10, 19,  asr->ASR in ref 18 …).

---

### Decision · Program_Chairs · 2023-06-14

**Decision:**

Accept

**Comment:**

SSW2003 received 45 papers. The acceptance rate is 82%. We are pleased to inform you that your paper has been accepted by the SSW2023 Program Committee. Please read the reviews carefully and submit your camera-ready paper by June 28th. Most of reviewers performed a detailed review. Please answer to their questions and take into account their comments.
Since your paper received a score below 5/9 that is strongly argued by the reviewers, note that the Program Committee will check if your manuscript has been significantly changed to specifically consider their remarks. Note that camera-ready papers are credited with one extra page to allow authors to consider reviewers’ suggestions. So max 7 pages in total including figures & refs.
The deadline for submitting the revised version (with full non anonymized authors and refs!) is 28th June.